# Utility of PROM Questionnaires: Correlation of Question Burden and Response Rate Among Surgically Treated Patients with Musculoskeletal Diseases

**DOI:** 10.3390/jcm14196728

**Published:** 2025-09-24

**Authors:** Karita Vilkki, Ville Äärimaa, Saara Meronen, Joel Kostensalo, Hanna-Stiina Taskinen, Ida Rantalaiho, Anssi Ryösä, Katri Pernaa, Inari Laaksonen

**Affiliations:** 1The Faculty of Medicine, University of Turku, Kiinamyllynkatu 4–8, 20521 Turku, Finland; ville.aarimaa@tyks.fi (V.Ä.); ida.rantalaiho@varha.fi (I.R.); inari.laaksonen@tyks.fi (I.L.); 2Department of Orthopedics and Traumatology, Turku University Hospital, Luolavuorenkatu 2, 20720 Turku, Finland; saara.meronen@tyks.fi (S.M.); hanna-stiina.taskinen@tyks.fi (H.-S.T.); anssi.ryosa@tyks.fi (A.R.); katri.pernaa@tyks.fi (K.P.); 3Natural Resources Institute Finland (Luke), Yliopistokatu 6B, 80100 Joensuu, Finland; joel.kostensalo@luke.fi

**Keywords:** elective operation, follow-up, institutional registry, motivation, musculoskeletal condition, patient-reported outcome measure (PROM), question burden, response rate

## Abstract

**Objectives:** Patient-reported outcome measures (PROMs) constitute a patient-centered way to assess treatment outcomes in musculoskeletal diseases. In this study, we explored the factors affecting the response rate when systematically utilizing PROMs for operatively treated patients in a clinical setting. The purpose was to find factors that could be influenced to improve the response rates of questionnaires in the future. **Methods**: The data were collected from a large institutional registry, divided into seven sub-registries (hand, elbow, shoulder, back, hip, knee, and foot and ankle), by gathering demographic data and joint-specific and generic PROM results. The data were collected preoperatively, as well as postoperatively at 3 months and 1 year. We analyzed patient demographics, the questionnaire format, and the length of each questionnaire, which were hypothesized to be the factors associated with the response rate. **Results**: The study sample consisted of 2295 patients with operatively treated musculoskeletal conditions. A response rate of 60% or above was obtained for the whole patient cohort at all three time points, although not in all sub-registries. A higher number of questionnaire items (−0.021, *p* < 0.001) and the patients’ smoking status (−0.395, *p* = 0.002) were associated with a lower response rate. The response rate increased with the patient age up to 75 years and decreased thereafter. **Conclusions**: A suitable limit for the number of questions in a PROM questionnaire might be 50 to ensure the required 60% response rate to obtain generalizable results. Special effort should be made to improve the PROM response rate among the younger adult patient population.

## 1. Introduction

Patient-reported outcome measures (PROMs) have become the conventional methods of evaluating health states, health state changes, and treatment outcomes, especially in non-life-threatening conditions, such as musculoskeletal diseases, and in elective orthopedic surgeries [1].

PROM questionnaires can be either generic, representing a patient’s overall quality of life, or specific, emphasizing a particular disease/condition or joint [1,2]. The development of disease-specific PROMs has enabled the detection of even small changes in health, which otherwise would remain unidentified by generic PROMs [3]. To obtain the best possible general perspective on the effectiveness of medical treatment from the patient’s point of view, generic and disease-specific PROMs are often combined.

To obtain representative and reliable results from a patient cohort, each PROM questionnaire’s response rate should be at a minimum of 60% [4,5]. All additional items in the questionnaire increase the question burden and potentially jeopardize patient compliance, the response rate, and completeness [6,7,8]. Monetary incentives, shorter questionnaires, prenotification, and follow-up notification, on the other hand, have been reported to improve response rates [8,9]. Furthermore, collecting PROM questionnaires both online and via mail produces significantly higher response rates than using the first option only [3].

However, there is limited knowledge on the patient demographic factors affecting response rates, as well as a lack of recommendations regarding the maximum number of questions in PROM questionnaires. While demographic factors cannot be altered, further understanding of which patients are less likely to respond is vital for developing demographically targeted approaches to increase the response rate. Furthermore, questionnaires are often collected both electronically and on paper, and understanding the differences in response rates between these formats is key to successful interventions for increasing the response rate.

In this study, our main aim was to analyze the association between the question burden and the response rate for each PROM questionnaire, based on a large institutional registry. Secondly, we aimed to investigate how the response rates of patients representing different demographics (e.g., age and sex) differed from each other, and thirdly, we aimed to evaluate the proportional use of online and mailed questionnaires.

## 2. Materials and Methods

An institution-based systematic PROM quality registry with an observational benchmark research objective (T283/2020) for all patients undergoing elective surgeries for musculoskeletal conditions was established in 2019 at Turku University Hospital, Turku, Finland. The registry was divided into sub-registries by specialties, based on a patient’s musculoskeletal condition (i.e., hand, elbow, shoulder, spine, hip, knee, and foot and ankle). From hand, shoulder, spine, and foot and ankle sub-registries, questionnaires on generic PROMs, preoperative patient history, and other data (e.g., preoperative) were collected in addition to questionnaires on joint-specific PROMs (Table 1). The joint-specific PROMs were always asked before the other questionnaires. The number of items in all questionnaires combined (i.e., the question burden) in the sub-registries was calculated (Table 1). The registry questionnaire survey was activated using an electronic application (Omavointi, BCB Medical, Turku, Finland) for each eligible patient who was booked for a preoperative assessment at a specialist outpatient clinic. The patients received an invitation letter by mail before their outpatient visit, including instructions for answering the questionnaires online. Furthermore, the hospital staff was instructed to encourage the patients to fill in the questionnaires during all care contact. The surveys were conducted at 3 months and at 1 year postoperatively. A link to each questionnaire was sent via SMS (short messaging service, text message) 2 weeks before each time point, and the patients received a follow-up reminder SMS if the scores were missing 2 weeks after the due date. If no response was received after the latter reminder, a paper questionnaire that asked the same questions and had an enclosed prepaid return envelope was mailed to the patients at 1 year postoperatively, but this only applied to the shoulder, spine, hip, and knee sub-registries.

The collected data varied among the sub-registries, and in addition to PROM data included age, sex, native language (Finnish, Swedish, Lapp, English, or other), occupational status (employed, unemployed, student, retired, or other working status), smoking status (non-smoking or smoking), other diseases (no diseases, cardiovascular disease, neurological disease, rheumatological disease, endocrinological disease, lung disease, renal disease, liver disease, cancer, psychiatric disease, allergy, or other disease), the American Society of Anesthesiology (ASA) score, and the body-mass index (BMI). Surgery-related data on the operation day, the main diagnosis code (according to the International Classification of Diseases (ICD)-10 classification), and the main operation (according to the Classification of Surgical Procedures Version 1.14 by the Nordic Medico-Statistical Committee [NOMESCO]) were also gathered.

The response rate (i.e., the proportions of respondents and non-respondents in each sub-registry) was calculated at each time point, including the patients whose preoperative questionnaires were activated and who were each scheduled for an elective surgery between 1 January and 30 June 2020. The patients who underwent operative treatment until 1 November, 2020, were included to ensure that a whole year of postoperative data collection was complete at the time of the data extraction. The elbow sub-registry was not launched until 2 March 2020, and due to the small number of patients (<30), it was excluded from our analyses. The patients who underwent a revision surgery during the 1-year follow-up period were also excluded because an early revision would disrupt the PROM collection time cycle.

### Statistics

The effect of four factors—age, physical status (as measured by the ASA classification system), sex, and the number of all questions in a questionnaire—on the probability of answering PROM questionnaires was studied as the primary research question. Based on graphical investigations of the data, the probability of answering depended on the age of the respondent, though not in a consistent way. Thus, a generalized additive mixed model (GAMM) [24,25], which allows the response to depend on age in an arbitrary and smooth way, was chosen to estimate the statistical significance of the factors. The ASA class, sex, and the number of questions were treated as fixed linear effects, age as a smooth fixed effect, and the respondent and the specific registry as random effects. Random effects are necessary to induce the covariance structure of the data because of people’s tendency for individual responses, and some sub-registry-specific effects may not be captured by the fixed effects. ASA classes were not available for the hand sub-registry, so they were left out of the GAMM fit. There were no missing data for the ASA class, age, sex, sub-registry, or the number of questions once the hand sub-registry was excluded. Observations with missing PROM data were excluded.

The secondary variables of interest were the smoking habits and employment status of each patient. These variables were investigated by adding them to the model as fixed effects, one at a time. However, it should be noted that these variables were missing from the records of approximately half of the patients; thus, the statistical power was significantly smaller than that in the main model, as all entries with missing smoking or employment status were excluded.

All calculations were carried out using the statistical software R (4.4.2) [26], and visualizations were performed using the package ggplot2 (4.4.0) [27]. The GAMM model was fitted using the R package mgcv (version 1.8-39). In addition, see Appendix A.

## 3. Results

Between 1 January and 30 June 2020, a registry follow-up with a PROM questionnaire was activated for 6727 patients. Of these patients, 2278 were operated on and had their preoperative PROM questionnaire completed before 1 November 2020. In total, 2295 cases were included in the analyses since some of the patients had undergone multiple different surgeries during the study period. The patients had a mean age of 60 years (SD = 16, range = 14–94), and 1325 (58%) were female. The most common ASA class was 2 (930 patients, 41% of all patients), followed by class 3 (693, 30%). Smoking status was recorded in 1145 patients (50%), of whom 25% of men and 15% of women smoked (Table 2).

The total numbers of respondents and non-respondents at any time point were 2034 (89%) and 261 (11%), respectively. The response rate was 60% or higher in the whole study population at all three time points (72% preoperatively, 60% at 3 months, and 63% at 1 year). The response rate at each time point for each sub-registry is presented in Table 3. In the hand, spine, and foot and ankle sub-registries, the response rate showed a statistically significant decrease (*p* < 0.001, *p* = 0.02, and *p* < 0.001, respectively) from the preoperative to the postoperative time points. Of these, the spine sub-registry still had a sufficient response rate at the postoperative time points (64%), whereas the hand and the foot and ankle sub-registries were clearly below 60% (36% and 41%, respectively).

An increase in the number of questions (question burden) had a statistically significant relation to a lower response rate (−0.021, *p* < 0.001) in a multivariable GAMM model (Figure 1, see Appendix A for full model details). The results can be summarized well by a rule of thumb, where the response rate is 0.99^N^, and N denotes the number of questions.

Age was significantly statistically associated with the response rate (*p* < 0.001); for patients up to 75 years old, the response rate increased, and after this, a decrease was observed (Figure 2). When only the online electronic responses were assessed, the highest response rate was found among 59-year-old patients at 61.4% (Figure 3). For a healthy (ASA class = 1) 60-year-old patient, a 60% response rate was achieved, with 50.8 questions. Sex, ASA class, and occupational status had no statistically significant relation to the response rate. Smoking was associated with a poorer response rate (−0.395, *p* = 0.002). (See Appendix A).

In total, 3042 (44%) of the responses were collected electronically and 1447 (21%) manually in paper format (Table 3).

## 4. Discussion

The main finding of this study was the significant negative association of question burden and response rates among patients surgically treated for musculoskeletal conditions. The overall PROM questionnaire response rate was satisfyingly 60% or higher preoperatively, as well as at 3 months and at 1 year postoperatively. However, in several sub-registries with a heavy question burden, the response rates were unsatisfactory.

Traditionally, orthopedic surgery results have been assessed with overly blunt tools, such as revision rates [4]. Moving toward more patient-centered and value-based thinking is vital for improving healthcare outcomes [1,28]. Using PROM data to evaluate treatment outcomes and effectiveness benefits all healthcare stakeholders [1]. PROMs enable comparisons of treatment outcomes at local and larger levels, improvement of clinician–patient communication as part of shared decision-making, facilitation of comparative effectiveness research, and enhancement of patients’ awareness of treatment outcomes [1,29]. However, to objectively evaluate results and ensure their generalizability, a sufficiently high response rate should be achieved. A minimum of 60% response rate as the threshold is often presented in the literature, and in cases of a poor response rate, both respondents and non-respondents should be surveyed to determine potential bias [1,4,5,30]. Therefore, the usefulness of clinical quality registries has been under a cloud of suspicion [31,32]. Our study demonstrates the general applicability of a systematic institution-based PROM registry with an overall satisfactory response rate.

It is challenging to strike a balance between long questionnaires that provide detailed information on the treated conditions and questionnaires that are short enough to achieve high response rates. In our study, the question burden seemed to be the factor with the clearest impact on the response rate. Other studies have reported similar results for both online and mailed questionnaires [8], where shorter questionnaires increase the response rates by over 50%. The foot and ankle sub-registry, which has the highest number of preoperative questions (75 questions) of all our sub-registries, showed that the odds of responding to the questionnaire plummeted statistically significantly from 72% preoperatively to 37% at 3 months and 35% at 1 year postoperatively. Presumably, after completing the lengthy questionnaire once, patients are less likely to fill out the same questionnaire again. The fact that the shortest questionnaires in our sub-registries (hip arthroplasty and knee arthroplasty, 12 questions each) show an increase in their response rates over time seems to support this assumption. However, in these sub-registries, at the 1-year time point, the assisting staff members send printed questionnaires to patients who have not submitted the completed online questionnaires, which likely has an effect on the final response rate. This approach was not used at 3 months, when these registers also show higher-than-average response rates. Nonetheless, this finding highlights the need for shortening our questionnaires to obtain response rates high enough to draw generalizable conclusions from the results. In our data, among the healthy 60-year-old patients, the required 60% response rate was achieved with 50 or fewer questions. Some potential practical strategies to achieve this goal in the future include could be: patient’s ability to answer only to the specific PROM (and not the entire PROM questionnaire), carefully assessing which questions are relevant in the entire PROM questionnaire in order to reduce the overall question burden, and highlighting the importance of the studies and the further benefit of the results obtained from them for patients.

According to our results, smoking was associated with a low response rate. Since several sub-registries did not record the patients’ smoking status, extrapolating these results to a larger population should be undertaken with caution. It should be noted that the model we used was fitted using only 50% of the data: those that included the smoking status. However, the percentage of smokers (18.6%) among those patients whose smoking status was documented in our material was close to the Finnish average of 12% [33]. In earlier studies, an active smoking status has also been associated with loss to follow-up among patients with spine disorders, which is in line with our findings [34,35]. This may be due to smokers’ worse preoperative disease states and larger number of comorbidities [34].

In our study, age and response rate showed a statistically significant correlation. The response rate increased among patients up to 75 years old and decreased thereafter. When the response rate to only electronic questionnaires was studied, the respondents were, on average, a year younger than the non-respondents. In a Dutch study, younger age was associated with a decreased response rate from preoperative to postoperative PROM questionnaires, consistent with our data, as their patient population was younger than 75 years [34]. Harris et al. reported that younger patients were more likely to fill out electronic questionnaires, but, with a follow-up telephone call, the difference by age diminished, and overall, the discrepancy was small [30]. Furthermore, Ling et al. found no correlation between age and response rate among shoulder arthroplasty patients [36]. However, both studies involved only arthroplasty patients, who were also older than those in some of our sub-registries (mean age = 67 years [SD = 9]).

Providing both online and paper options for answering PROM questionnaires increases the response rate [36,37]. Contacts from a research assistant have been reported to increase response rates by 20% [38]. In our material, approximately two out of three patients answered online, and one-third filled out a paper questionnaire. Sending questionnaires by mail is time-consuming and costly; however, currently, we cannot achieve a sufficiently high response rate with only an electronic approach. In our institutional registry, systematic monitoring of whether the patient has answered the questionnaires at the 1-year time point is carried out only with the shoulder, spine, hip, and knee sub-registries. In these sub-registries, the paper questionnaires are then sent if the electronic response is missing. This factor contributes to the postoperative response rate of these sub-registries. In addition, it is important to critically consider whether the answers to electronic questionnaires are really that reliable: electronic answering can make it easier to give hasty answers compared to paper forms. The patient may also become frustrated if the electronic questionnaire gets stuck/is poorly developed. It would also be interesting to know what kind of impact different individuals’ digital literacy has on the likelihood of responding to electronic surveys. On the other hand, some studies have found that PROM responses obtained using the electronic method have been similar to those obtained using the traditional paper questionnaire [39,40]. In addition, electronic questionnaires have the potential to reach a significantly larger number of patients more cost-effectively compared to paper forms.

We acknowledge our study’s limitations. First, we studied neither the psychometric properties of the PROMs nor the treatment results. Instead, we focused only on the number of questions, patient demographics, and response rates. The PROMs used in the sub-registries were selected by our sub-specialists who specialized in musculoskeletal diseases of the joint in that particular sub-register; therefore, they contained varying numbers of questions. Second, due to some variance in the data among the different sub-registries, we were unable to include all of them in all analyses. Third, some of the sub-registries had relatively low numbers of patients. Fourth, the patients in all sub-registries received SMS reminders if they had not answered the postoperative questionnaires, but, in some sub-registries, the study nurse also directly contacted the non-respondents, which might have affected the response rates. Approximately 5500 elective musculoskeletal surgeries are performed annually at Turku University Hospital. The discrepancy between the actual number of annual surgeries and the smaller number of patients included in our study is another limitation, which may be partly caused by the failure to activate preoperative questionnaires for some patients (e.g., due to an emergency or direct admission to surgical treatment). Furthermore, 1–2% of the patients refused to fill out the questionnaires and participate in the PROM registry.

The strengths of this study lie in the large patient population and the systematic PROM collection system in a large orthopedic unit covering a population of 500,000 people. The large institutional registry’s highly detailed and accurate data enable further high-quality research. The PROMs used in our registry are also validated, and the same PROMs are used in several different countries, which increases the possibility of generalizing the results. On the other hand, the results may not be as generalizable to environments where several different hospitals operate in a small area, possibly relying on multiple information systems.

## 5. Conclusions

There is a strong correlation between shorter patient questionnaires and higher response rates. To achieve a sufficient response rate, the number of questions should be limited, both in the beginning and during the follow-up.

Special effort should be made to improve the response rate among younger adults. Finally, providing both electronic and mailed response options and ensuring that the assisting staff is large enough to collect the completed questionnaires are necessary to obtain sufficient response rates.

## Figures and Tables

**Figure 1 jcm-14-06728-f001:**
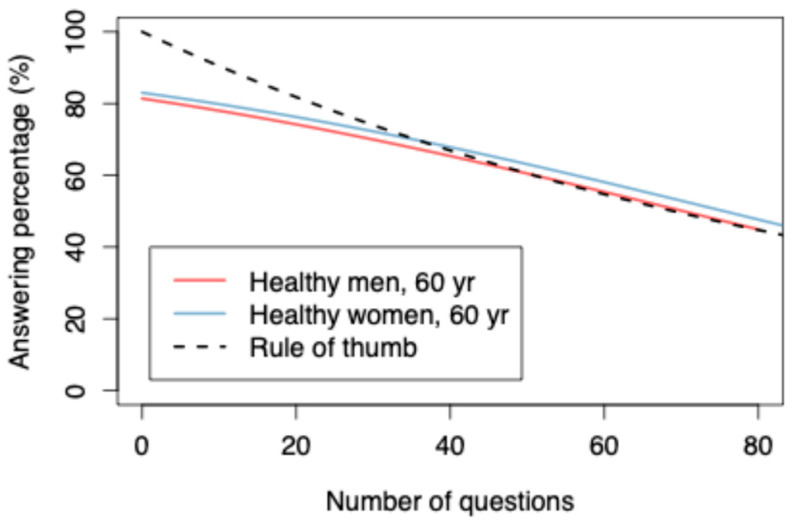
The correlation between the response rate and the number of questions.

**Figure 2 jcm-14-06728-f002:**
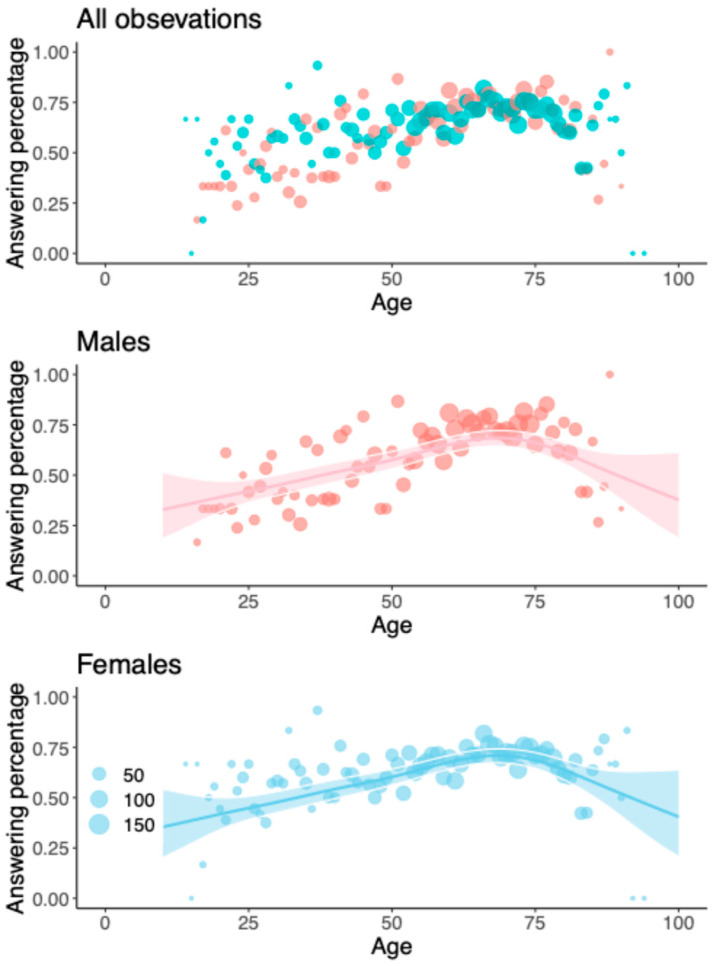
Response rates for the whole study population and males and females separately. The red dots represent males and the blue ones females. The lines and the shaded areas represent GAMM predictions and their 95% confidence intervals for a typical registry (random intercept zero) with 48 questions and ASA class I.

**Figure 3 jcm-14-06728-f003:**
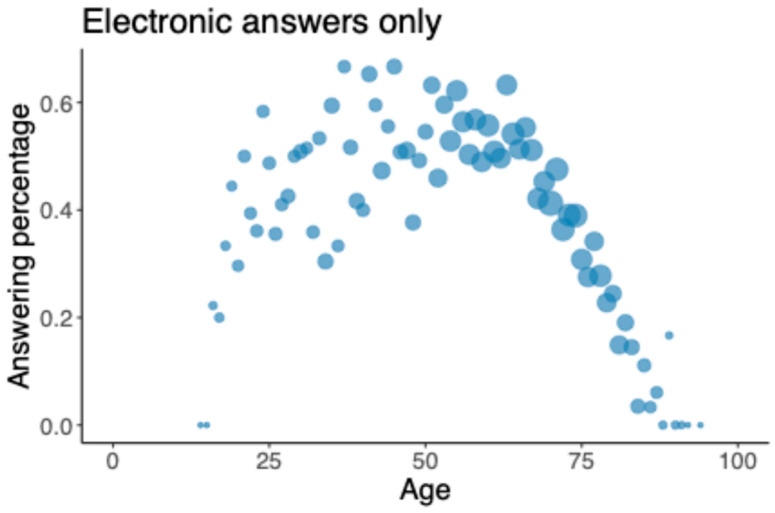
Response rate for online questionnaire only (including all patients).

**Table 1 jcm-14-06728-t001:** The sub-registries in the institutional PROM registry; the collected questionnaires for specific PROMs, generic PROMs, and other data; and the preoperative and postoperative question burden.

Sub-Registry		Specific PROM	Generic PROM	Other Questionnaires	Preoperative Questions (N)	Postoperative Questions (N)
Hand		QUICKDASH	-	Preoperative data, PCS, follow-up form	45	33
Shoulder						
	Instability	WOSI	15-D	Preoperative data, SSV	54	37
	Rotator cuff	WORC	15-D	Preoperative data, SSV	54	37
	Arthroplasty	WOOS	15-D	Preoperative data, SSV	52	35
Spine		ODI/NDI	EQ-5D	Preoperative data, VAS, surgical satisfaction	38	31
Hip						
	Arthroplasty	OHS	-	-	12	12
	Other	OHS	-	-	12	12
Knee						
	Arthroplasty	OKS	-	-	12	12
	Other	KOOS	-	-	42	42
Foot and ankle		FAOS	15-D	Preoperative data, SFAV	75	58

PROM = patient-reported outcome measure, WOSI = Western Ontario shoulder instability [10], WORC = Western Ontario rotator cuff [11], WOOS = Western Ontario osteoarthritis of the shoulder [12], OHS = Oxford hip score [13], OKS = Oxford knee score [14], KOOS = knee injury and osteoarthritis outcome score [15], FAOS = foot and ankle outcome score [16], 15-D = 15-dimensional measure of health-related quality of life [17], EQ-5D = EuroQol-5D—a health-related quality-of-life questionnaire [18], ODI = Oswestry disability index [19], NDI = neck disability index [19], QUICKDASH = quick disabilities of arm, shoulder, and hand [20], SSV = subjective shoulder value [21], SFAV = subjective foot/ankle value, VAS = visual analog scale [22], PCS = pain catastrophizing scale [23].

**Table 2 jcm-14-06728-t002:** Demographic data on all patients in the sub-registries.

		All	Hand	Shoulder			Spine	Hip		Knee		Foot and Ankle
				Instability	Rotator Cuff	Arthroplasty		Arthroplasty	Other	Arthroplasty	Other	
N		2295	614	15	49	39	600	317	17	411	64	169
Mean age (range)		60 (14–94)	54 (15–94)	34 (16–62)	60 (34–90)	71 (48–86)	59 (17–89)	69 (22–91)	39 (21–71)	69 (28–91)	35 (14–70)	55 (17–83)
Female, *n* (%)		1325 (58)	370 (60)	6 (40)	15 (31)	24 (62)	313 (52)	198 (62)	9 (53)	243 (59)	23 (36)	124 (73)
Smoking, *n* (%)												
	Yes	214 (9)	112 (18)	2 (13)	8 (16)	6 (15)	69 (12)	-	-	-	-	17 (10)
	No	931 (41)	326 (53)	9 (60)	29 (59)	26 (67)	417 (70)	-	-	-	-	124 (73)
	NA/missing	1150 (50)	176 (29)	4 (27)	12 (24)	7 (18)	114 (19)	317 (100)	17 (100)	411 (100)	64 (100)	28 (17)
ASA, *n* (%)												
	I	394 (17)	107 (17)	9 (60)	8 (16)	3 (8)	101 (17)	34 (11)	8 (47)	34 (8)	41 (64)	49 (29)
	II	930 (41)	168 (27)	5 (33)	34 (69)	16 (41)	248 (41)	158 (50)	6 (35)	185 (45)	23 (36)	87 (51)
	III	693 (30)	81 (13)	1 (7)	7 (14)	18 (46)	236 (39)	124 (39)	3 (18)	191 (46)	0 (0)	32 (19)
	IV	25 (1)	5 (0.8)	0 (0)	0 (0)	2 (5)	15 (3)	1 (0.3)	0 (0)	1 (0.2)	0 (0)	1 (0.6)
	NA/missing	253 (11)	253 (41)	0 (0)	0 (0)	0 (0)	0 (0)	0 (0)	0 (0)	0 (0)	0 (0)	0 (0)

ASA = American scale of anesthesiology.

**Table 3 jcm-14-06728-t003:** Response rates in total and in the sub-registries, required reminders, and the response methods used.

		All	Hand	Shoulder			Spine	Hip		Knee		Foot and Ankle
				Instability	Rotator Cuff	Arthroplasty		Arthroplasty	Other	Arthroplasty	Other	
Response rate for the specific PROM (%)												
	Preoperative	72	67	60	73	72	71	78	47	78	64	72
	3 mo, postoperatively	60	47	33	69	67	56	81	29	82	41	37
	1 yr, postoperatively	63	34	40	67	79	71	89	47	89	53	35
Response received after (%):												
	1st SMS reminder	33	29	29	45	32	38	32	29	33	38	37
	2nd SMS reminder	10	13	9	12	13	10	6	10	6	10	16
Response method (%)												
	Electronic	44	43	40	58	45	49	41	41	42	48	53
	Preoperative	51	59	60	59	49	47	44	47	46	59	54
	3 mo, postoperatively	39	60	33	65	46	49	30	29	33	39	47
	1 yr, postoperatively	42	31	27	49	41	52	50	47	45	47	54
	Paper	21	7	7	16	31	18	42	0	42	5	10
	Preoperative	21	9	7	18	26	26	35	0	33	6	17
	3 mo, postoperatively	21	8	0	6	26	9	51	0	50	2	6
	1 yr, postoperatively	21	3	13	22	41	21	39	0	44	6	8

mo = months, yr = year, SMS = short messaging service, text message.

## Data Availability

The datasets presented in this article are not readily available because hospital data contains patient identification data, and GDPR prevents us from sharing individual raw data. Requests to access the datasets should be directed to the corresponding author.

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
