# Peer review of "Utility of PROM Questionnaires: Correlation of Question Burden and Response Rate Among Surgically Treated Patients with Musculoskeletal Diseases"

_jcm, 2025, doi:10.3390/jcm14196728_

Round 1
Reviewer 1 Report
Comments and Suggestions for Authors
The title and abstract have insufficient clarity and precision.
The introduction does not define the knowledge gap offering statements rather than a critical synthesis of existing evidence on PROM response factors.
Methodologically, the study is missing data on smoking, heterogeneity across sub-registries with very different questionnaire lengths, limited adjustment for confounders and insufficient transparency about the statistical modeling process. Please revise
Results are adequately presented.
The discussion tends to overgeneralize - recommending a maximum of 50 questions despite the heterogeneity of the data and presents conclusions about smoking that are stronger than warranted given the missing information...I recommend a full recheck
Important alternative explanations for low response rates, such as digital literacy or socioeconomic status are not considered and the selective use of mailed questionnaires in only some sub-registries is not examined and/or presented
The references include outdated sources with limited use of recent literature on digital PROM collection
Author Response
Author's Reply to the Review Report (Reviewer 1)
Thank you for your comments and time spent on our article. We have tried to provide an answer to all of your concerns, and the detailed responses can be found below. All changes made to the manuscript have been highlighted with red font in the revised manuscript.
Comments and Suggestions for Authors
The title and abstract have insufficient clarity and precision.
Response 1: Thank you for this comment, we have now refined the title to be more informative and to match better with the abstract.
The introduction does not define the knowledge gap offering statements rather than a critical synthesis of existing evidence on PROM response factors.
Response 2: Thank you for this attention. We had previously addressed these shortcomings (limited knowledge on the patient demographic factors affecting response rates, lack of recommendations regarding the maximum number of questions) in the text related to the topic, but they were overshadowed by the paragraph that was too long. We have now shortened the paragraph to make it clearer and to make these issues better visible to the reader.
Methodologically, the study is missing data on smoking, heterogeneity across sub-registries with very different questionnaire lengths, limited adjustment for confounders and insufficient transparency about the statistical modeling process. Please revise
Response 3: Thank you for this comment. Indeed, in the original version of the manuscript the presence/absence of missing data was not explicitly stated. We have now revised Section 2.1 by adding the sentences “There were no missing data for ASA class, age, sex, sub-registry, or number of questions once the hand sub-registry was excluded. Observations with missing PROM data were excluded” to stress that the confounders that we controlled for did not include any missing data. We have also added the remark “…as all entries with missing smoking and employment status were excluded.” to Section 2.1 to highlight that smoking status was not imputed. We would like to stress, that not only do we control for several fixed effects, like age, ASA class, and number of questions, but the registry-specific random effect included in the GAMM is designed to deal with the inevitable heterogeneity in the data caused by the investigation of different anatomical parts.
Results are adequately presented.
Response 4: Thank you for your comment, it’s great to hear that results -paragraph has been clear and relevant.
The discussion tends to overgeneralize - recommending a maximum of 50 questions despite the heterogeneity of the data and presents conclusions about smoking that are stronger than warranted given the missing information...I recommend a full recheck
Response 5: Thank you for your careful attention. Reviewer is right here, there are many confounding factors in our study. However according to our study and data 50 question was the limit to achieve the required 60 % response rate. We have now made changes to our article so that the recommendation is not quite as strict and radical (lines 31-33, 316-317).
We have indicated in several places in the text (lines 144-145, 159-160, 248-256) that data on smoking were only available in some sub-registers and for this reason the results should be viewed critically. However, our data included responses from over a thousand patients regarding smoking, and in this patient population, the prevalence of smoking was quite close to overall Finnish smokers, which again supports the generalizability of the findings.
Important alternative explanations for low response rates, such as digital literacy or socioeconomic status are not considered and the selective use of mailed questionnaires in only some sub-registries is not examined and/or presented
Response 6: Our registry also includes data of occupational status (employed, unemployed, student, retired, other working status; lines 106-107). In our original text, the issue had been discussed and reviewed in the results, but we had decided to remove it from the text to streamline the article, because this information was also only available in some sub-registers and the end result was that it had no statistically significant impact on response coverage. We have now added this topic back to the manuscript.
We have now also taken into account digital literacy briefly in the discussion section.
The sub-registries were originally separate from each other and were created by different specialists (this is mentioned in the text, lines 291-294). As a result, the PROM questionnaires differ in their entirety and there may also be other practical differences in the sub-registries, for example in the methods of collecting responses that could not be influenced at the time of conducting the study.
The references include outdated sources with limited use of recent literature on digital PROM collection
Response 7: Thank you for this notification. We tried to do a little searching for newer references, but several new publications from 2024-2025 had originally referred to older material. However, we have now added a few newer references to the text in the introduction and discussion sections.
Reviewer 2 Report
Comments and Suggestions for Authors
The authors present a valuable study exploring the relationship between question burden and response rates in PROMs among surgically treated musculoskeletal patients. The manuscript is overall well-written, methodologically sound, and addresses an important issue for outcome research. My specific comments are as follows:
1. The exact method of handling these missing values (e.g., exclusion, imputation) should be more explicitly stated in the Methods section. This will strengthen the transparency of the analysis.
2. For the limitation, the author need to clarify the limitation for online-based research.
3. The authors rightly acknowledge limitations in terms of single-institution data. It would be beneficial to expand briefly on how findings might (or might not) generalize to other healthcare systems or international settings, given differences in PROM adoption and registry infrastructures.
4. The conclusion emphasizes reducing question burden below 50 items. It would add value if the authors could also suggest potential practical strategies (e.g., adaptive questionnaires, prioritization of core items) to achieve this goal without sacrificing clinical relevance.
Author Response
Author's Reply to the Review Report (Reviewer 2)
Thank you for your comments and time spent on our article. We have tried to provide an answer to all of your concerns, and the detailed responses can be found below. All changes made to the manuscript have been highlighted with red font in the revised manuscript.
Comments and Suggestions for Authors
The authors present a valuable study exploring the relationship between question burden and response rates in PROMs among surgically treated musculoskeletal patients. The manuscript is overall well-written, methodologically sound, and addresses an important issue for outcome research. My specific comments are as follows:
- The exact method of handling these missing values (e.g., exclusion, imputation) should be more explicitly stated in the Methods section. This will strengthen the transparency of the analysis.
Response 1: Thank you for pointing this out. There were no missing data in the main model once the hand sub-registry was removed, which we have now explicitly stated in Section 2.1 with the added sentences “There were no missing data for ASA class, age, sex, sub-registry, or number of questions once the hand sub-registry was excluded. Observations with missing PROM data were excluded.” Furthermore, we now explicitly state that observations with missing smoking status were excluded from the second model (see the clarification “…as all entries with missing smoking and employment status were excluded.” added to Section 2.1).
- For the limitation, the author need to clarify the limitation for online-based research.
Response 2: Thank you for this notification. We added some perspectives of this topic to the revised manuscript (see lines 279-288).
- The authors rightly acknowledge limitations in terms of single-institution data. It would be beneficial to expand briefly on how findings might (or might not) generalize to other healthcare systems or international settings, given differences in PROM adoption and registry infrastructures.
Response 3: Thanks for your comment. We have expanded the topic at the end of the discussion section, taking into account your recommendations (lines 307-313).
- The conclusion emphasizes reducing question burden below 50 items. It would add value if the authors could also suggest potential practical strategies (e.g., adaptive questionnaires, prioritization of core items) to achieve this goal without sacrificing clinical relevance.
Response 4: That is a very important point of view. We have now added some thoughts of strategies in the discussion chapter that might help with this goal in the future (see lines 242-247).
Of course there are also different PROMIS questionnaires that bring their own unique approach to the topic.
Round 2
Reviewer 1 Report
Comments and Suggestions for Authors
All the comments I raised were addressed.